# Thermal and Mechanical Characterization of Yarn Samples from Flemish Tapestry of the Sixteenth Century

**DOI:** 10.3390/molecules27238450

**Published:** 2022-12-02

**Authors:** Maria Rita Caruso, Lorenzo Lisuzzo, Giuseppe Cavallaro, Giacomo Mirto, Stefana Milioto, Giuseppe Lazzara

**Affiliations:** 1Dipartimento di Fisica e Chimica Emilio Segrè, Università degli Studi di Palermo, Viale delle Scienze, pad. 17, 90128 Palermo, Italy; 2Laurea Magistrale in Conservazione e Restauro per i Beni Culturali, Università degli Studi di Palermo, Viale delle Scienze pad 17, 90128 Palermo, Italy; 3Laurea Magistrale in Conservazione e Restauro per i Beni Culturali, Università degli Studi di Palermo, Scancarello srl, Palermo, Via Ugo Betti, 3, 90147 Palermo, Italy

**Keywords:** yarn samples, thermogravimetry, dynamic mechanical analysis, water uptake

## Abstract

We propose a physico-chemical approach for theharacterization of the conservation condition of yarns from a Flemish tapestry of the sixteenth century. The aging effect on the yarns’ performance was evaluated by comparison with commercial materials. Water uptake experiments highlighted the aptitude of yarns toward water sorption and their increased hydrophilicity upon aging. Thermogravimetric analysis can be considered a fast approach for the fiber identification and assessment on the material life-time. The dynamic mechanical analysis provided direct evidence on the yarns, conservation state and their performance under different mechanical stresses. The proposed characterization path can be relevant for stating the condition of the tapestry and for designing a conservation protocol for the preservation of the artwork.

## 1. Introduction

Among cultural heritage artifacts requiring a conservation operation, biopolymeric-based materials such as wood and textiles are very common due to the availability and suitability for decorative elements such as decorative objects, tapestries, carpets, and prestigious clothes. Moreover, due to aging effects, textile-based natural fibers often need a proper treatment protocol to improve or restore their features [1,2,3,4,5,6,7].

Scientists and conservators have to work to identify the conservation condition of these artworks, and with this aim an advanced physico-chemical approach is required to plan further restoration actions, as are the best conditions possible to prevent the artwork from further damage [8,9,10,11,12,13].

A proper approach for the objective diagnostic of the object can be achieved only by measuring quantitatively useful physico-chemical parameters. In fact, besides any visible alteration, the mechanical strength, response to humidity, and thermal stability are some of the key aspects that should be taken into account during display and for a proper design of a storage environment with the aim of preventive conservation.

This article is about the characterization of yarn samples from a Flemish tapestry of the Sixteenth Century from an unknown artist. These goods belong to the Tapestry Museum of Marsala in Sicily (Italy). The tapestry depicts the Judaic-Roman war and, in particular, scenes from Vespasian’s military campaigns against the Jews. This tapestry is made of wool-based warp and silk-based weft. Wool is the most commonly used animal fiber as a weft material for fabrics and is obtained from the soft, hairy covering of sheep and sometimes goats. Silk, another common animal fiber, is made by the mulberry silkworm when spinning its cocoon and, being a strong continuous filament yarn, it was the ideal warp yarn for expensive fabrics of this type. 

Ageing effects in wool and silk are mainly due to alterations of the respective protein chemistries. It has been reported that UV radiation can have a crucial role due to the photochemical sensitivity of histidine, tryptophan, tyrosine, among the least stable amino acids [14].

The HPLC–MS procedure was successfully used in determining the complex composition of yarns collected from a 15th century tapestry and their conservation conditions [14]. Advanced spectroscopic methods have been employed for the characterization of yarns with a particular focus on the pigments [15,16] or influence of the fiber surface as a consequence of detergent residue adsorption [13]. As reported in the literature, characterization of wool samples and historical threads is also carried out by Double Shot Py-GC/MS and EGA/MS, which highlight protein and lipid fractions of wool [1]. Moreover, EGA/MS is suitable for quickly assessing the conservation conditions of the woolen yarns [1].

Despite the great interest in the cultural heritage field, to our knowledge the quantitative assessment of the physico-chemical parameters providing a bulk response of the tapestry upon stress due to environmental conditions (humidity and temperature) and mechanical performances of fabric as a consequence of aging are not always fully addressed. With this work, we have demonstrated the relevance of a physico-chemical characterization by dynamic mechanical analysis, thermogravimetry (conventional and modulated) and water uptake experiments as excellent analytical methods for the evaluation of the conservation conditions and material identification. This approach can be considered a key characterization for the preventive conservation and identification of the safe storage conditions for aged textiles of this type.

## 2. Results and Discussion

The study involved a mechanical characterization (tensile tests and dynamic-mechanical experiments under an oscillatory regime) implemented by water uptake measurements (moisture adsorption) and thermal stability (TG and modulated TG) on ancient yarn samples. The physico-chemical properties of the analyzed yarns were compared with the corresponding characteristics of commercial merino wool and “Botto” silk yarns for warp and weft references, respectively. The optical microscopy images for the warp showed the wool-typical aspect of a long cylinder with surface scales on it as well as having a high crimp. On the other hand, the silk weft filaments appear to be smooth and shiny cylinders.

### 2.1. Water Uptake 

The commercial merino wool and the “Botto” silk samples were subjected to water uptake investigations to assess their tendency to absorb water vapor and, consequently, their hygroscopicity. The measurements were conducted by exposing the samples to a controlled atmosphere (temperature of 25 °C and relative humidity of 75%). The exposure time in the controlled atmosphere was systematically varied with the aim of obtaining kinetic information on the water absorption process. Figure 1 shows the water uptake values at different time (5, 10 and 15 days) for each sample. It should be noted that results after 30 days did not show any relevant difference compared to the water uptake after 15 days (See Appendix A). This result agrees with the literature findings providing ca. 15 days of equilibration time for similar yarn samples [17].

### 2.2. Tensile and Dynamic Mechanical Analysis

Yarns in their different application processes are often required to support loads, either in a static manner (simple tensile) or in a dynamic one (fatigue). The effect of both types of stress have been tested in this work by DMA measurements as reported elsewhere for similar samples [18,19].

The yarns were subjected to tensile tests to determine their tensile properties by analyzing the corresponding stress vs strain curves (Figure 2). The measurements were carried out in isothermal conditions (temperature of 25 °C) and after equilibration with a relative humidity of 55%.

Figure 2 clearly shows that all the wool-based warps analyzed have tensile properties inferior to those of the commercial merino wool. The parameters obtained for the tensile properties, and in particular the elastic modulus, the stress at the breaking point, the maximum elongation, are shown in Table 1 for a better comparison. As concerns the silk-based yarns, one can conclude that the aging does not have the strong effect on the mechanical performance observed for the wool samples. This result agrees with the recent report on the low aging impact on the mechanical performance of silk weft [20]. Contrarily, it should be noted that the stress at breaking is reduced by a factor of four at least in the wool-based samples, implying a particular care while loading is applied in the warp direction.

It was found that the lowest-performing yarn is the brown warp, which provided the lowest breaking stress and elastic modulus. 

Nevertheless, although the tensile performance of the aged yarns are inferior to the commercial yarn, they still have sufficient mechanical strength to withstand the stresses required for exhibiting them in a vertical position, although monitoring any further tensile property deterioration should be considered.

Dynamic-mechanical measurements in oscillatory mode were carried out with the aim of investigating their rheological properties and the fatigue response of the material. It has been observed that an increase in temperature leads to a reduction of all the rheological parameters with a monotonic trend (examples are provided in the Appendix A). The same trends were found for merino wool, “Botto” silk and for the other yarns, highlighting that the viscoelastic response to temperature variations is similar in all the samples analyzed. 

Table 2 reports the values of the rheological parameters obtained at 25 °C. The values of the viscoelastic modules (storage modulus and loss modulus) of all the wool-based yarns analyzed are lower than those of commercial merino wool. As regards tan (δ) it was found that blue and red warp have lower values than merino wool, while an opposite result was observed for brown warp. The yellow weft has a value of tan (δ) similar to that of commercial silk. Finally, the rheological modules evidenced that the warp yarns are more damaged by aging than the silk samples, which do not show a relevant change in performances.

### 2.3. Thermogravimetric Analysis

Thermogravimetric analysis represents a powerful, low-cost and micro-destructive method for the investigation of artworks [2,21,22,23]. Measurements carried out under modulated heating ramp conditions can be used to determine the activation energy for the degradation process, which is sensitive to the state of conservation and also can be a useful tool to predict the lifetime of a material [24]. Mass loss profiles (Figure 3) showed an onset of the main degradation step at ca. 230 °C for wool samples (both aged and commercial merino sample) whilst it was at 280 °C for silk and weft samples (Table 3). These results agree with the reported TG profile differences between silk and wool [25,26] and therefore confirm the possibility of distinguishing between these two yarns by a relatively easy and fast measurement. Water content in the yarns was estimated by the mass loss at 150 °C, and it provides values in agreement with the water uptake experiments (Table 3). The residual mass after the pyrolytic process does not show a systematic variation that can be used for the material characterization (Table 3). As concerns the activation energy values for the pyrolytic process, they were calculated by modulated thermogravimetry (MTG) experiments in all conversion degree ranges by using the method reported in the literature [27]. Examples of raw data are provided in Appendix A; average activation energy values in Table 3 indicate that the aged samples still have a comparable value with the commercial samples and therefore that the yarns are in a good conservation condition from the chemical point of view. It should be noted that the activation energy is strongly dependent on the degradation of a material. For instance, for wood samples it significantly increases in aged samples due to the absence of the most degradable portion with lower activation energy [2,28].

## 3. Materials and Methods

### 3.1. Materials

Figure 4 reports the optical image of the Flemish Tapestry and the micrographs of the investigated yarns. It should be noted that the weft is made on silk whilst the warp yarns are wool-based. Therefore, the comparison with non-aged samples was carried out by measuring a “Botto” silk and Merino wool commercial samples, respectively.

### 3.2. Methods

#### 3.2.1. Thermogravimetric Analysis (TG)

Thermogravimetric (TG) analyses were carried out on a Q5000 IR apparatus (TA Instruments, Vimodrone. Milan, Italy) under nitrogen atmosphere (gas flows of 25 and 10 cm 3 min^−1^ were employed for the sample and the balance, respectively). The experiments were carried out by heating the sample (ca. 2 mg) at 20 °C min^−1^. Modulated TGA data were collected on a TGA 550 (TA Instruments, Vimodrone. Milan, Italy) apparatus. The sample (ca. 10 mg) was heated up at an average heating rate of 2 °C min^−1^ with an oscillation amplitude of 5 °C and a period of 200 s. Water content was calculated from the mass loss at 150 °C.

#### 3.2.2. Dynamic Mechanical Analysis (DMA)

Dynamic Mechanical Analysis (DMA) was conducted by a DMA Q800 apparatus (TA Instruments, Vimodrone. Milan, Italy) following two different protocols: stress vs strain at fixed temperature and dynamic stress under a temperature ramp. For the isothermal experiments, a stress ramp of 10 MPa min^−1^ at 25.0 ± 0.5 °C was used. We determined the mechanical performances in terms of stress at breaking (σr), deformation at breaking (σr) and elastic modulus (EM). The latter was calculated from the slope of the linear stress vs strain curves. The mechanical response to the temperature was conducted in the oscillatory regime (frequency of 1.0 Hz and a stress amplitude of 20 MPa) by heating the samples to 70 °C at a heating rate of 3 °C min^−1^. In all cases the yarn diameter was determined by a micrometer (±10–3 mm).

#### 3.2.3. Water Uptake (WU)

The water uptake (WU) experiments were done on the samples that were dried under vacuum at 25 °C for ca. 2 h until constant weight. After weighing, they were conditioned at 75% relative humidity (RH%) in a climate chamber containing saturated salt solutions of NaCl (Sigma product). The temperature of the climate chamber was 25.0 ± 0.5 °C. The samples were weighed (±0.00001 g) until constant weight. The WU% was calculated as:WU% = 100 · (Mt − M0)/Mt(1)
where M0 and Mt are the weights of the sample before and after equilibration.

#### 3.2.4. Optical Microscopy (OM)

Optical images were produced by using a DIGITUSs (DA-70351) microscope (ASSMANN Electronic GmbHpe, Lüdenscheid, Germany) and processed by using the Digital Viewer 5.7 software.

## 4. Conclusions

The present study has demonstrated the potential success of less conventional methods for the characterization of aged yarns. The quantitative evaluation of physico-chemical parameters relating to bulk features such as water sorption capability, mechanical strength and thermal resistance have a dual relevance: firstly, the methodology has produced a robust approach to identify the damage and conservation condition; and secondly, it provides insights on the actual environmental and exposition features to be addressed for preservation of the tapestry.

Further research will be focused on a more general multidisciplinary approach that can be used to draw better conservation protocols from the quantitative evaluation of bulk properties in a wide range of biopolymeric-based artworks.

## Figures and Tables

**Figure 1 molecules-27-08450-f001:**
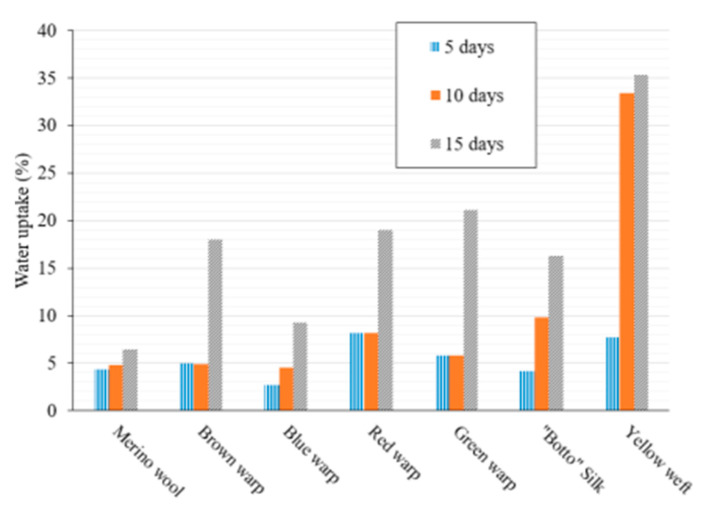
Water uptake values of commercial Merino wool, of “Botto” silk and of the Sixteenth Century yarns (brown, blue, red and green wrap and yellow weft yarns). The results were obtained at 25 °C and relative humidity of 75%.

**Figure 2 molecules-27-08450-f002:**
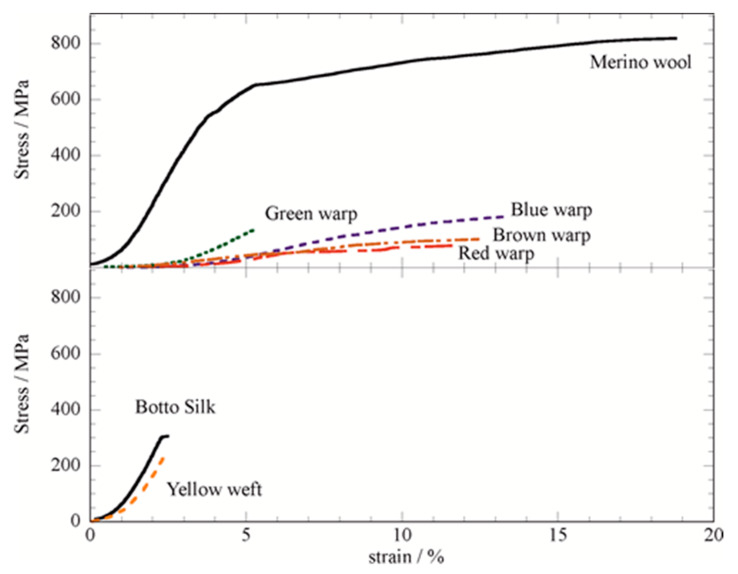
Stress vs. strain of commercial Merino wool, “Botto” silk and the Sixteenth Century yarns.

**Figure 3 molecules-27-08450-f003:**
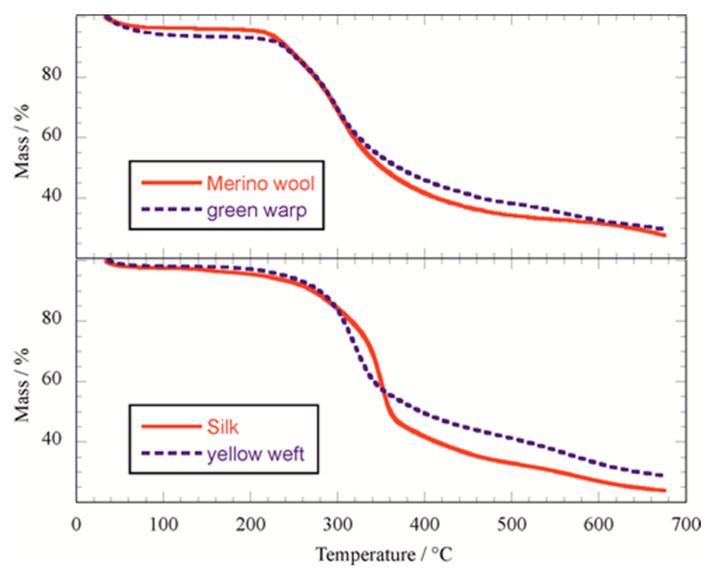
Thermoanalytical curves of Merino wool, “Botto” silk and the Sixteenth Century yarns.

**Figure 4 molecules-27-08450-f004:**
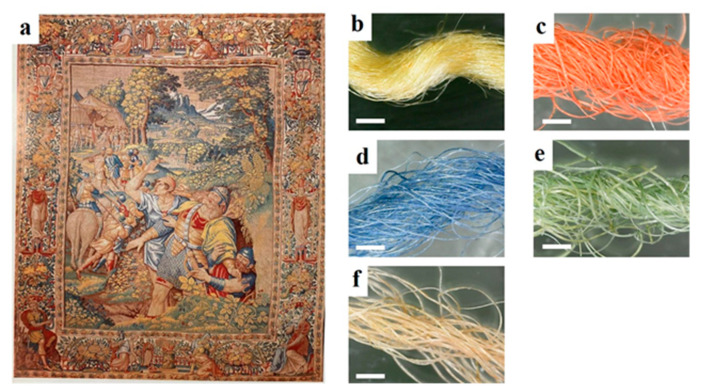
Optical photo of the Flemish Tapestries (348.5 cm × 250 cm) of the Sixteenth Century (**a**) and optical micrographs of yarns: yellow weft (**b**), red warp (**c**), blue warp (**d**), green warp (**e**) and brown warp (**f**). Scale bars are 1 mm.

**Table 1 molecules-27-08450-t001:** Tensile properties at 25 °C of commercial Merino wool, “Botto” silk and the Sixteenth Century yarns.

	Elastic Modulus (GPa)	Stress at Breaking (MPa)	Maximum Elongation (%)
Merino wool	19.6 ± 1.4	815 ± 57	25.9 ± 1.8
Blue warp	2.43 ± 0.18	178 ± 13	13.2 ± 1.0
Red warp	1.68 ± 0.12	78 ± 6	11.6 ± 1.2
Green warp	4.7 ± 0.4	139 ± 12	5.3 ± 0.4
Brown warp	1.22 ± 0.10	101 ± 9	12.4 ± 1.3
“Botto” Silk	10.2 ± 0.8	305 ± 22	2.52 ± 0.15
Yellow weft	15.7 ± 0.9	238 ± 18	2.41 ± 0.15

**Table 2 molecules-27-08450-t002:** Rheological properties at 25 °C of commercial merino wool, Botto silk and Sixteenth Century yarns.

	Storage Modulus (MPa)	Loss Modulus (MPa)	Tan (δ)
Merino wool	41.1	8.6	0.209
Blue warp	8.35	0.96	0.115
Red warp	15.1	2.0	0.136
Brown warp	25.9	5.9	0.228
“Botto” Silk	18.2	3.4	0.198
Yellow weft	15.5	3.2	0.214

Error on experimental data is 5%.

**Table 3 molecules-27-08450-t003:** Activation energy for the degradation process under inert atmosphere of commercial Merlin wool, “Botto” silk and the Sixteenth Century yarns.

	Activation Energy (kJ mol^−1^)	Water Content (wt%)	Residual Mass at 675 °C (wt%)	T_onset_ (°C)
Merino wool	260 ± 25	3.8	27.7	228.2
Green warp	250 ± 21	6.6	29.1	229.3
“Botto” Silk	270 ± 27	2.8	23.8	278.9
Yellow weft	260 ± 24	2.2	28.8	279.0

## Data Availability

The data presented in this study are available in article or Appendix A.

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
