# Peer review of "Thermal and Mechanical Characterization of Yarn Samples from Flemish Tapestry of the Sixteenth Century"

_molecules, 2022, doi:10.3390/molecules27238450_

Round 1

Reviewer 1 Report

Dear Editor,

The paper “Thermal and Mechanical Characterization of Yarn samples 2 from Flemish Tapestry of the Sixteenth Century” by Maria Rita Caruso , Lorenzo Lisuzzo , Giuseppe Cavallaro, Giacomo Mirto, Stefana Milioto and Giuseppe Lazzara reports a physico-chemical characterization of yarn samples from historical manufact. The approach appears correct and the results worth of publication. The comparison with contemporary yarns provided further information’s that are valuable for a molecular understanding of the system.

I suggest minor revisions as follows:

- Authors should check units as superscripts are not in the proper place. See for instance Methods section.

- Symbol for “deformation at breaking” is missed. See line 169.

- Errors for tensile properties in table 1 should be included for a proper comparison.

- About TG data analysis, it would be interesting to add a table with residual mass, water content, degradation temperature (onset, for instance) for a more quantitative description/comparison.

- Lines 78-79 “Figure 1 shows the water uptake values at different time (5, 10 and 15 days) for each sample. It should be noted that results after 30 days did not show any relevant difference compared to the water uptake after 15 days.” But Data after 30 days are not present in figure 1, please add.

- The sentence (lines 147-149) “This section may be divided by subheadings. It should provide a concise and precise 147 description of the experimental results, their interpretation, as well as the experimental conclusions that can be drawn.“ must be deleted

- It has been reported in literature that differential and integral isoconversional methods gave same results when properly used https://doi.org/10.1007/s10973-015-4741-7 but the graphs of the obtained activation energies by MTG have to be reported at least in the supplementary materials

- The paper https://doi.org/10.1016/j.jaap.2018.09.012 can be added and commented in the introduction.

Best regards,

Author Response

We acknowledge the Reviewer for his/her positive opinion on our paper and for giving us the opportunity to improve the quality of the manuscript.  The replies to all the comments/suggestions of the Reviewer are reported as follows.

- Authors should check units as superscripts are not in the proper place. See for instance Methods section.

- The text has been revised.

- Symbol for “deformation at breaking” is missed. See line 169.

- The symbol (σr) for “deformation at breaking” has been added.

- Errors for tensile properties in table 1 should be included for a proper comparison.

- Errors for tensile properties have been added in Table 1.

- About TG data analysis, it would be interesting to add a table with residual mass, water content, degradation temperature (onset, for instance) for a more quantitative description/comparison.

- Thermogravimetric parameters (water content, residual mass at 675 °C and Degradation temperature from onset point) have been added in Table 3.  

- Lines 78-79 “Figure 1 shows the water uptake values at different time (5, 10 and 15 days) for each sample. It should be noted that results after 30 days did not show any relevant difference compared to the water uptake after 15 days.” But Data after 30 days are not present in figure 1, please add.

- Water uptake values after 30 days have been added in Supplementary Materials.

- The sentence (lines 147-149) “This section may be divided by subheadings. It should provide a concise and precise 147 description of the experimental results, their interpretation, as well as the experimental conclusions that can be drawn.“ must be deleted

- The sentence has been deleted in the revised MS.

- It has been reported in literature that differential and integral isoconversional methods gave same results when properly used https://doi.org/10.1007/s10973-015-4741-7 but the graphs of the obtained activation energies by MTG have to be reported at least in the supplementary materials.

- As example, the activation energy as a function of the conversion degree obtained from MTG for botto silk sample has been added in Supplementary Materials.  

- The paper https://doi.org/10.1016/j.jaap.2018.09.012 can be added and commented in the introduction.

- The suggested reference has been added and commented in the Introduction.

Reviewer 2 Report

This is an interesting short paper, which with improvements regarding my annotated comments, should be considered for publication.

Author Response

We acknowledge the Reviewer for his/her positive opinion on our paper and for giving us the opportunity to improve the quality of the manuscript. All the suggestions and comments of the Reviewer have been considered in the revised MS.   

Round 2

Reviewer 2 Report

The authors have addressed my comments. In line 146, you introduce the term "MTG" which obviously refers to modulated TG. Please refer to the term in full since you have not defined it in section 3.2.1.

Author Response

We thank the reviewer for his/her careful work. 

In line 146, we have added the full term "modulated thermogravimetry" for MTG.